# A Facile Strategy for Fabrication Lysozyme-Loaded Mesoporous Silica Nanotubes from Electrospun Silk Fibroin Nanofiber Templates

**DOI:** 10.3390/molecules26041073

**Published:** 2021-02-18

**Authors:** Jingxin Zhu, Haijuan Wu, Ding Wang, Yanlong Ma, Lan Jia

**Affiliations:** 1College of Materials Science and Engineering, Taiyuan University of Technology, Taiyuan 030024, China; wuhaijuan0093@link.tyut.edu.cn (H.W.); mayanlong@tyut.edu.cn (Y.M.); jialan@tyut.edu.cn (L.J.); 2BOE Photoelectricity Technology Co., Ltd., Chengdu 611731, China; wangdingtyut@126.com

**Keywords:** mesoporous silica nanotubes, electrospinning, silk fibroin, drug release, lysozyme

## Abstract

This paper presents a facile and low-cost strategy for fabrication lysozyme-loaded mesoporous silica nanotubes (MSNTs) by using silk fibroin (SF) nanofiber templates. The “top-down method” was adopted to dissolve degummed silk in CaCl_2_/ formic acid (FA) solvent, and the solution containing SF nanofibrils was used for electrospinning to prepare SF nanofiber templates. As SF contains a large number of -OH, -NH_2_ and -COOH groups, the silica layer could be easily formed on its surface by the Söber sol-gel method without adding any surfactant or coupling agent. After calcination, the MSNTs were obtained with inner diameters about 200 nm, the wall thickness ranges from 37 ± 2 nm to 66 ± 3 nm and the Brunauer–Emmett–Teller (BET) specific surface area was up to 200.48 m^2^/g, the pore volume was 1.109 cm^3^/g. By loading lysozyme, the MSNTs exhibited relatively high drug encapsulation efficiency up to 31.82% and an excellent long-term sustained release in 360 h (15 days). These results suggest that the MSNTs with the hierarchical structure of mesoporous and macroporous will be a promising carrier for applications in biomacromolecular drug delivery systems.

## 1. Introduction

Hollow-structured mesoporous silica nanospheres and nanotubes, which have large void space, high specific surface area, good thermal stability, and biocompatibility, and easy surface modification, have attracted great attention in adsorption and separation, storage, catalysis, and drug delivery systems [1,2,3,4]. So far, the drug delivery system based on mesoporous silica nanospheres has been greatly developed [5,6,7,8,9]. However, mesoporous silica spheres have smaller bore diameter (<5 nm), which are limited in application for the loading and diffusion of biomacromolecular drugs, such as DNA/RNA, enzymes, and proteins. The mesoporous silica nanotubes (MSNTs) are one-dimensional nanotubes with anisotropy and a large aspect ratio, and have a hierarchical structure of mesoporous and macroporous. Compared with other kinds of nanotubes, such as carbon nanotubes and titania nanotubes, silica nanotubes are not only more easily fabricated via an ambient and low-cost sol-gel route, but also exhibit better biocompatibility, which have been confirmed in vitro and in vivo cytotoxicity tests [10,11], Therefore, MSNTs may be interesting and useful carrier for biomacromolecular drugs.

For the fabrication of MSNTs, hard-template strategy is more prevailing due to the fact that MSNTs with highly controllable hollow core diameter, shell thickness, and mesoporosity in the shell. Various templates, such as cylindrical polymer brushes [12], carbon nanotubes [13], and surfactant and nanocrystalline cellulose [14], have been applied to determine the final nanostructure of MSNTs. However, the synthesis of these templates is complex, high-cost, and difficult to scale up.

Electrospinning is a simple and efficient process to produce continuous fibers with diameters ranging from micrometers to nanofibers (NFs). By using the fibers as templates, MSNTs can be fabricated via sol–gel route and calcination process. For example, Wang et al. prepared TiO_2_/silica nanotubes, with a high specific area and high photocatalytic activity, by combining electrospinning technique, sol–gel process and the calcinations [15]. Jia et al. prepared hierarchical MSNTs by sol-gel process with the aid of the surfactant (cetyl trimethyl ammonium bromide (CTAB)) using electrospun cellulose acetate fibers as sacrifice templates [16]. Guo et al. explored the use of electrospun gelatin nanofibers as templates to produce MSNTs and evaluated their drug-loading properties [17].

Silk fibroin (SF) of Bombyx mori, the major component of silkworm silk fibers, is an abundant and low-cost resource in China. SF, composed of 18 amino acids, is rich in amino and carboxyl groups in its molecules, making SF a good template for the preparation of MSNTs. To produce SF nanofiber templates, an electrospinning solution need to be prepared, and the general method is to dissolve silk by the chaotropic salt system, such as Ca(NO_3_)_2_‧4H_2_O/CH_3_OH, LiBr/H_2_O, CaCl_2_/C_2_H_5_OH/H_2_O, etc. [18,19,20,21]. However, these dissolution systems generally require a long time of dialysis desalination process (about 3 days), which reduces the preparation efficiency. Zhang et al. [22] reported that CaCl_2_/formic acid (FA) solvent could rapidly dissolve silk into the nanoscale, and the SF solution, which containing nanofibrils, could be used for the preparation of high-quality silk fibroin materials [23,24,25,26,27,28,29].

Lysozyme is a naturally macromolecular enzyme. Lysozyme degrades peptidoglycans present in the bacterial cell wall which causes cell death [30]. Lysozyme has antibacterial properties against gram-positive bacteria. It can be used to prevent bacterial infection and promote the wound healing [31,32]. Lysozyme has also been a promising candidate in the application of pharmacological functions. Herein, the aim of this study was to developed a simple, highly efficient, and low-cost method for the fabrication biomacromolecular drug-loaded MSNTs. Initially, the “top-down method” was adopted to dissolve degummed silk in CaCl_2_/ FA solvent, and the solution containing nanofibrils was used for direct electrospinning to produce SF nanofiber templates. Then, by adopting Söber sol-gel method, silica could be easy induced to nucleate and deposit on the surface of SF nanofibers due to a large number of –OH, –NH_2_, and –COOH groups in SF molecules [33]. After calcination, the MSNTs were obtained and their properties were characterized. Finally, lysozyme was selected as a bioactive protein drug to be loaded on MSNTs. The results suggested that the MSNTs prepared according to our method exhibited relatively high drug encapsulation efficiency and an excellent long-term sustained release behavior compared with traditional mesoporous silica spheres, they will be preferred silica–based carrier for applications in biomacromolecules drugs delivery system.

## 2. Results and Discussion

### 2.1. The Viscosity of SF/CaCl_2_/FA Solutions

The viscosity of the solution is closely related to the spinnability of the spinning dope and is also an indicator for protein aggregates structure in the solution. The use of CaCl_2_/FA solvent for silk dissolution could peel native silk, which composed of multilevel fibrils with diameters in the range from 20 to 170 nm [34,35], from the micron fibers to the nanofibrils. The diameter and the length of the nanofibrils were affected by the content of CaCl_2_ and SF in SF/CaCl_2_/FA system and strongly determining the viscosity of the spinning dopes [24]. Our preliminary experiments also showed that when the SF was 15 (*w/v*) % and the content of CaCl_2_ was less than 3 (*w/v*) %, the dissolution rate of silk fibers was smaller and the viscosity of the solution was lower, which induced the jet of electrospinning was instable and obtained beaded-like fibers with uneven diameter.

Meanwhile, when the content of CaCl_2_ is 5 (*w/v*) % and the SF is greater than 25 (*w/v*) %, the viscosity of solution was higher; the electrospun fibers had a ribbon-like shape and the nanofibers adhesive together. Figure 1a shows the viscosity changes of SF/CaCl_2_/FA solutions with the content of CaCl_2_ when SF was concentration 15 (*w/v*) %. It can be seen that the viscosity of the solutions increased from 0.45 to 0.78 Pa‧S when the concentration of CaCl_2_ in CaCl_2_/FA solvent increased from 3 to 8 (*w/v*) %. This is due to the size and the diameter of silk nanofibrils decreased in SF/CaCl_2_/FA solutions (SEM images were shown in the Appendix A), inducing the solution viscosity increased with the increase of the interaction between nanostructure. Figure 1b shows the viscosity changes of SF/CaCl_2_/FA solution with the concentration of SF when CaCl_2_ was concentration 5 (*w/v*) %. It is noted that when SF concentration increased from 15 to 25 (*w/v*) %, the viscosity of the solutions increased from 0.55 to 2.0 Pa‧S. Moreover, when SF concentration reached more than 20 (*w/v*) %, the viscosity increased significantly. This is because with the amount of SF molecules and nanofibrils in the SF/CaCl_2_/FA solution increased, the entanglement and interaction between SF molecules and nanofibrils increased, thus the viscosity of the solution increased. In addition, Figure 1 shows that the viscosity of SF/CaCl_2_/FA solutions reached a stable state between 2–6 h after a slight increase, which provided ample time for electrospinning.

### 2.2. Morphology of Electrospun SF Nanofiber Templates

The morphology of the electrospun SF nanofibers was observed by SEM, as shown in Figure 2. The surface of the electrospun fibers was relatively smooth. With the increase of CaCl_2_ concentration from 3 (*w/v*) % to 8 (*w/v*) % when SF concentration was 15 (*w/v*) %, the average diameter of SF fibers increased from 145 ± 29 nm to 241 ± 55 nm (Figure 2a–c. On the other hand, with the increase of SF concentration from 15 (*w/v*) % to 25 (*w/v*) % when CaCl_2_ concentration was 5 (*w/v*) %, the average diameter of SF fibers increased from 158 ± 47 nm to 344 ± 99 nm (Figure 2d–f). These results indicated that with the increase of the viscosity of spinning dopes, the diameters of the fibers increased. In other words, the diameters of the fibers were closely related to the amount of silk nanofibrils and the proteins aggregation state in the spinning dopes. By adjusting the concentration of CaCl_2_ and SF in the spinning dopes, the diameters of the fiber templates could be easily regulated.

### 2.3. Morphologies and Compositions of SF@silica NFs and MSNTs

SF electrospinning fibers contain a large number of –OH, –NH_2_, and –COOH on their surface, and the presence of Ca^2+^ in the fibers make the surface of the fiber have good hydrophilicity, which provides a great possibility for inducing hydrolysis tetraethoxysilane (TEOS) to condense on its surface [33,36,37]. By soaking SF fiber templates in Söber-type sol-gel solution, the silica shell was successfully formed on the surface of SF fibers without adding any surfactant or coupling agent. After calcination, the hollow-structured mesoporous silicon nanotubes (MSNTs) could be obtained by removing the SF fiber templates from silica-coated SF nanofibers (SF@silica NFs). Figure 3 shows SEM images and EDS spectrum of SF@silica NFs before and after calcination, the SF@silica NFs prepared from the SF fiber templates with the concentration of CaCl_2_ at 3, 5, and 8 (*w/v*) % when SF concentration was 15 (*w/v*) %, respectively. It can be seen that the coating effect on the surface of the fiber templates was basically the same although the diameters of the fiber templates were different (Figure 3a–c). After calcination (Figure 3e–g), the MSNTs retained the form of the templates, the silica particles on the surface were closely arranged and evenly distributed, and the cross-section of MSNTs were in good shape without collapse or break. Figure 3d,h show the EDS spectrum of SF@silica NFs and MSNTs. Before calcination (Figure 3d), there were Si elements in SF@silica NFs in addition to C, O, and N elements of SF fibers, which demonstrated that the surface of the SF fibers was coated with silica particles. After calcination (Figure 3h), there were only Si and O elements remaining in the chemical composition of the MSNTs, and the mass fraction of Si elements was greatly increased compared with that before calcination, which indicates the basic removal of SF templates.

### 2.4. FTIR Spectra of SF Fiber Templates, SF@silica NFs and MSNTs

Figure 4 shows FTIR spectra of SF fiber templates, SF@silica NFs, and MSNTs. The FTIR spectra of SF fiber templates after treated with methanol (Figure 4a) exhibited the peaks at 1633 cm^−1^ and 1520 cm^−1^, corresponding to the amides I and amide Ⅱ characteristic peaks of β-sheet structures in SF fibers [38,39]. Besides the characteristic peaks of SF fibers, the FTIR spectra of SF@silica NFs (Figure 4b) exhibited all the characteristic peaks of silica. The peak at 1100 cm^−1^ assigned to the antisymmetric stretching vibration of Si-O-Si. Positions of 804 and 463 cm^−1^ were assigned to the symmetric stretching vibration and the bending vibration of Si-O, respectively, meanwhile, the peak at 954 cm^−1^ corresponding to the bending vibration of Si-OH [16,40]. After calcination, the FTIR spectra of MSNTs only shows the characteristic peak of silica, the peaks at amides Ⅰ and amide Ⅱ of SF templates and the peak of Si–OH all disappeared, indicating that the SF templates were removed and the Si-O-Si was formed between the Si-OH by removing the water molecules.

### 2.5. Effect of TEOS Dosages Changes on the Silica Layer Thickness of MSNTs

TEOS is the silicon source for preparation SF@silica NFs, so the addition of TEOS is directly relevant to the silica layer thickness of SF@silica NFs and MSNTs. In this experiment, we kept most of the composition of the coating solution at constant, such as aqueous ammonia, the volumes of water/ethanol, while varied TEOS addition from 3, 5 to 7 mL to investigate the effect of TEOS dosages on the silica layer thickness of MSNTs, and the three kinds of MSNTs were denoted MSNTs-1, MSNTs-2, and MSNTs-3. The TEM images are presented in Figure 5a–c. From these images, we can observe that the MSNTs had good shape, obvious the contrast and meso–microporous hierarchical structure. With the increase of TEOS addition, the wall surface of the MSNTs changed from relatively smooth to rough, and the wall thickness of the MSNTs increased gradually. According to the measurement, the average wall thickness of the MSNTs increased from 37 ± 2 nm, 46 ± 2 nm to 66 ± 3 nm. This suggested that the wall thickness of MSNTs can be controlled by the dosages of TEOS in the coating solution.

### 2.6. Thermogravimetric Analysis (TGA) of SF@silica NFs

TGA was used to test the influence of TEOS dosages on the residual mass of SF@silica NFs after the removal of SF fiber templates. The results are shown in Figure 6. It can be seen that there was little mass loss of SF fibers and SF@silica NFs before 250 °C. When the temperature rose to about 275 °C, the mass loss of SF fibers (Figure 6a) was significantly increased due to the decomposition temperature of the SF templates around 280 °C [41]. After the temperature rose to 600 °C, the SF residual mass was almost zero, indicating that the SF templates had been removed, which was consistent with the FTIR analysis (Figure 4), meanwhile, the residual mass of different SF@silica NFs (Figure 6b–d) was 43.3, 45.7, and 50.4 wt%, respectively, corresponding to the gradual increase of TEOS dosages. This result indicated that the higher the amount of TEOS added, the more silica deposited on the surface of SF fibers, which was in good agreement with the results shown in TEM images (Figure 5) and further confirmed that the TEOS dosages could adjust the wall thickness of MSNTs.

### 2.7. MSNTs Characterized by Brunauer–Emmett–Teller (BET)

N_2_ adsorption–desorption measurements were employed to determine the specific surface areas and pore structure of the obtained MSNTs-1, MSNTs-2, and MSNTs-3 (Figure 7). All of the samples exhibited capillary condensation and P/P_0_ from 0.3 to 1. The hysteresis loop was observed at relative pressures (P/P_0_) ranging from 0.3 to 0.85, indicating the presence of mesopores. The sharp capillary condensation that occurred at relative pressures (P/P_0_) between 0.85 and 1 could be explained by the presence of macropores, which might originate from the inner MSNTs hollow structure. The adsorption–desorption isotherm would be assigned to a Type IV isotherm according to the 1985 IUPAC (International Union of Pure and Applied Chemistry) [42], N_2_ adsorption–desorption curves provided robust pieces of evidence of the existence of hierarchical structures in the MSNTs. This result was consistent with the TEM observed (Figure 5).

The BET surface area of MSNTs-1, MSNTs-2, and MSNTs-3 was 200.48, 159.87, and 94.87 m^2^/g, respectively, after calculated by the BET method. The Barrett–Joyner–Halenda (BJH) pore size distributions of MSNTs-1, MSNTs-2, and MSNTs-3 were inserted in Figure 7, and the average mesopores diameter in walls of the MSMTs was 5.49, 5.97, and 5.75 nm respectively. The total pore volume of MSNTs-1, MSNTs-2, and MSNTs-3 was 1.109, 0.696, and 0.611 cm^3^/g, respectively.

### 2.8. Drug-Loading and In Vitro Release Behavior of MSNTs

In order to study the drug-loading efficiency of MSNTs for biomacromolecular drugs, we selected lysozyme as a model protein drug for the drug loading experiment. The results are shown in Table 1. It can be seen that the drug-loading efficiency and encapsulation efficiency of MSNTs were relatively high, in addition, with the decrease of the specific surface area and pore volume of the MSNTs from 200.48 to 94.87, the drug-loading efficiency decreased from 9.82 to 8.04 wt%, and the encapsulation efficiency decreased from 31.82 to 25.74 wt%. That is, with the decrease of the specific surface area and pore volume of the MSNTs, the adsorption capacity of the drug decreased slightly.

Figure 8 shows the release profiles of lysozyme from different lysozyme-loaded MSNTs (lysozyme-loaded MSNTs-1, lysozyme-loaded MSNTs-2 and lysozyme-loaded MSNTs-3 named as MSNTs-1-LYS, MSNTs-2-LYS, MSNTs-3-LYS). The release of lysozyme from the lysozyme-loaded MSNTs was divided into three stages: a burst release in the first 2 h, a moderate release between 2 and 270 h and stable slow release after 270 h. In the first stage, it could be due to the presence of some amount of lysozyme on the surface of the MSNTs, resulting in fast diffusion and release into the medium. In the second stage, lysozyme could diffuse from macropores and mesopores of lysozyme-loaded MSNTs, resulting in a slight increase in release; in the last stage, it might be that lysozyme could only diffuse from the mesoporous wall of the MSNTs, so the release entered the stable slow release stage. When the release reached up 360 h (15 days), the maximum release amount of MSNTS-1-LYS, MSNTS-2-LYS, and MSNTS-3-LYS were 63.0, 57.4, to 35.6%, respectively, due to the decrease of the specific surface area of MSNTs and the interactions between hydrophilic silanol groups (Si–OH) of silica nanotubes and hydrogen atom of lysozyme driven by hydrogen bonds. These results indicated that hierarchical structure lysozyme-loaded MSNTs had an excellent long-term sustained release effect compared with traditional mesoporous silica spheres [43].

## 3. Materials and Methods

### 3.1. Materials

Cocoons of *Bombyx mori* were purchased from Zhejiang, China. Calcium chloride (CaCl_2_‧6H_2_O) and lysozyme were purchased from Sangon Biotech Co., Ltd. Shanghai, China. Tetraethoxysilane (TEOS) were purchased from Shanghai Aladdin Bio-Chem Technology Co., Ltd., Shanghai, China. Formic acid (FA, >98%), sodium carbonate, methanol, ethanol, ammonia (28%) were purchased from Sinopharm Group Chemical Reagent Co. LTD. Shanghai, China. 

### 3.2. Preparation of Spinning Dopes

Cocoons were separated into pieces and boiled for 30 min with 0.5 wt% Na_2_CO_3_ solution, and washed thoroughly with deionized (DI) water to remove the residual sericin, repeat the above operation once more. After drying overnight, the degummed silk were dissolved in CaCl_2_/FA solvent with CaCl_2_ concentration of 3, 5 and 8 (*w/v*) % at 20 °C for 2.5 h yielding 15, 20 and 25 (*w/v*) % SF solutions containing SF nanofibrils (Scheme 1a). These SF/CaCl_2_/FA dopes were used for electrospinning.

### 3.3. Electrospinning for SF Nanofiber Templates

The electrospinning process is presented in Scheme 1b: the spinning dope was transferred to the 10 mL syringe equipped with a steel needle as a nozzle (0.55 mm in internal diameter). The syringe was mounted on a horizontal string pump (RWD 202, RWD Life Science Co., Ltd., Shenzhen, China). A voltage of 20 KV was applied for electrospinning with a flow efficiency of 0.5 mL/h and a distance of 15 cm between the needle tip and the collector. The environmental temperature and the humidity were 25 ± 2 °C, and 40 ± 5%, respectively. After the electrospinning, the SF nanofiber membranes were removed from the aluminum foil and then immersed in 90 (*V/V*) % methanol solutions for 20 min to induce formation transition for further stabilization.

### 3.4. Fabrication of Mesoporous Silica Nanotubes (MSNTs)

The fabrication of MSNTs is presented in Scheme 1c: SF nanofibers were soaked in the Söber-type sol-gel precursor solution, consisting of ethanol (37 mL), DI water (5 mL), and aqueous ammonia (28%, 0.3 mL), TEOS (3, 5, 7 mL, respectively), to produce silica-coated SF composite nanofibers (SF@silica NFs) at room temperature. After a reaction of 12 h, SF@silica NFs were washed twice with ethanol and DI water and calcined at 600 °C for 3 h to remove SF templates, then the mesoporous silica nanotubes (MSNTs) were obtained.

### 3.5. Drug Loading and Release Studies

Lysozyme was chosen as a biomacromolecules drug to study the drug loading and release property of MSNTs. For lysozyme loading, 20 mg of MSNTs were soaked in 10 mL of lysozyme solution (1 mg/mL) and the mixture was shaken at 100 rpm at 37 °C. After 12 h, lysozyme-loaded MSNTs were collected by centrifugation at 12,000 rpm for 30 min to remove the excess lysozyme. The supernatant was collected to determine the drug loading by an SQ-2800 UV-vis (UNICO, Shanghai, China) spectroscopy at the wavelength of 281 nm. The quality of lysozyme-loaded in MSNTs was calculated according to the established calibration curve (in the Appendix A
Appendix A). The drug-loading efficiency and encapsulation efficiency were calculated according to the following Equations (1) and (2):(1)$$\mathrm{drug}-\mathrm{loading}\;\mathrm{efficiency}(\mathrm{wt}\%)=\frac{\mathrm{the}\;\mathrm{quality}\;\mathrm{of}\;\mathrm{lysozyme}\;\mathrm{in}\;\mathrm{the}\;\mathrm{MSNTs}}{\mathrm{the}\;\mathrm{quality}\;\mathrm{of}\;\mathrm{the}\;\mathrm{lysozyme}-\mathrm{loaded}\;\mathrm{MSNTs}}\times 100\%$$
(2)$$\mathrm{encapsulation}\;\mathrm{efficiency}(\mathrm{wt}\%)=\frac{\mathrm{the}\;\mathrm{quality}\;\mathrm{of}\;\mathrm{lysozyme}\;\mathrm{in}\;\mathrm{the}\;\mathrm{MSNTs}}{\mathrm{the}\;\mathrm{quality}\;\mathrm{of}\;\mathrm{the}\;\mathrm{initial}\;\mathrm{lysozyme}\;\mathrm{addition}}\times 100\%$$

For in vitro drug release test, 11 mg lysozyme-loaded MSNTs samples were immersed in a tube with 10 mL PBS(Phosphate Buffered Saline)buffer (pH = 7.4) were shaken at 60 rpm at 37 °C. At appropriate intervals, its release medium (2 mL) was taken and immediately replaced with an equal volume of fresh. The cumulative drug release was calculated according to Equation (S1) (in the Appendix A, Equation (S1)).

### 3.6. Characterizations

Viscosity measurement of SF/CaCl_2_/FA solutions was used by a DV-II+PRO rotary viscometer (AMETEK Brookfield, Middleboro, MA, USA) at 20 °C. The morphology and the element content analysis of the samples were carried out using a Mira 3 scanning electron microscopy (SEM) equipped with energy-dispersive spectroscopy (EDS) detector (TESCAN Ltd., Brno, Czech). Fourier transform infrared spectrometer (FTIR) were recorded on a Tensor Ⅱ Spectrometer (Bruker, Germany) with a resolution of 4 cm^−1^ over the wavenumber range from 400–4000 cm^−1^. Transmission electron microscopy (TEM) analysis was conducted on a JEOL 2010 (JEOL Ltd., Tokyo, Japan) electron microscope at 200 kV. Thermogravimetric analysis (TGA) was conducted using a TG-209 F3 thermogravimetric analyzer (NETZSCH, Scientific Instruments Trading Ltd., Berlin, Germany) at a heating rate of 10 °C /min in a nitrogen atmosphere. Nitrogen adsorption/desorption isotherms were measured at 77 K with a JW-BK122T-B gas adsorption analyzer (Beijing JWGB Sci & Tech Ltd., Beijing, China) after the sample was first degassed at 120 °C for 3 h. The specific surface area was calculated by the Brunauer–Emmett–Teller (BET) method, the pore size distributions and the total pore volume were analyzed by the Barrett–Joyner–Halenda (BJH) method.

## 4. Conclusions

In summary, we developed a simple, highly efficient, and low-cost strategy for fabrication lysozyme-loaded hierarchical structure MSNTs by using SF electrospun fibers templates. Due to SF contains a large number of –OH, –NH_2_, and –COOH groups, silica could be easy induced to nucleate and deposit on its surface by Söber sol-gel method without adding any surfactant or coupling agent. After calcination, the uniform MSNTs were prepared with inner diameters about 200 nm, the wall thickness ranges from 37 ± 2 nm to 66 ± 3 nm and the BET specific surface area was up to 200.48 m^2^/g, the pore volume was 1.109 cm^3^/g. By using lysozyme as a model protein drug for drug loading, the MSNTs exhibited relatively high drug encapsulation efficiency up to 31.82% and an excellent long-term sustained release in 360 h (15 days). It is concluded that the MSNTs prepared by our strategy will be a promising carrier for applications in biomacromolecular drug delivery systems.

## Data Availability

Data is contained within the article or Appendix A.

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
