# Peer review of "A Facile Strategy for Fabrication Lysozyme-Loaded Mesoporous Silica Nanotubes from Electrospun Silk Fibroin Nanofiber Templates"

_molecules, 2021, doi:10.3390/molecules26041073_

Round 1
Reviewer 1 Report
for comments see attached file.

Author Response
Dear reviewers, thank you very much for your insightful comments and for answering the attachments I have uploaded.

Reviewer 2 Report
The submitted article is focused for the fabrication of lysozyme-loaded mesoporous silica nanotubes by using silk fibroin nanofiber templates. Prepared materials are presented as a promising carriers for the drug delivery system. Introduction is concisely and clearly worded and newer references has been used.The preparation and analysis process are well illustrated and discussed. After eliminating formal errors (typos, unify graphs on Figure 7), I recommend publishing this manuscript.
Author Response
We would like to thank the reviewer for encouraging comments and we are very sorry for our formal errors. We have examined the manuscript carefully and corrected the spelling and picture errors.
Reviewer 3 Report
Manuscript of Zhu et al is focused on fabrication of mesopore hollow nanotubes for sustainable drug release. This topic is recently of significant interest and importance. Submitted paper concerns original outputs especially from the point of material science.
I have additional questions and comments.
- Authors should state in Introduction reason why lysozyme was chosen as model biomacromolecular drug and emphasize the significance and role of lysozyme in therapeutic strategies.
- Authors should add in Introduction information concerning the current state of utilization of nanotubes as drug release systems.
- lines 245 ... Release of lysozyme provided only 25 to 55 % of total amount of the drug. Is there expected any interaction between silica particles and lysozyme?
- Authors provided calculation of encapsulation capacity of lysozyme , however they fixed the amount of lysozyme in the solution what was leading to moderate encapsulation capacity. Why authors did not provided the encapsulation of lysozyme at different concentration of lysozyme solution?
- Authors stated utilization of mesopore nanoparticles as drug delivery system. Therefore, I miss information about in vitro cytotoxicity effects of MSNTs, their internalization in cells and other effect on cell viability.
Minor comments:
- Authors should be consistent in utilization biomacromolecules drugs or biomacromolecular drugs.
- Authors should define all abbreviations in the first use (CTAB, BET, TEOS, EDS)
- Figure 2. Authors shoul add information about SF content (a-c) and CaCl2 content (d-f)
- line 170 MSNTs had good shape, this meaning is not clear, should be more informative
Author Response

(The authors gave the same response as above.)
